# Accuracy of the Infectious Diseases Society of America and British Thoracic Society Criteria for Acute Pneumonia in Differentiating Chemical and Bacterial Complications of Aspiration in Comatose Ventilated Patients Following Drug Poisoning

**DOI:** 10.3390/antibiotics13060495

**Published:** 2024-05-27

**Authors:** Quentin Delforge, Alexandre Gaudet, Pauline Boddaert, Frédéric Wallet, Benoit Voisin, Saad Nseir

**Affiliations:** 1Pôle Médecine Intensive-Réanimation, Hôpital Roger Salengro, CHU de Lille, 59000 Lille, Francealexandre.gaudet@chu-lille.fr (A.G.);; 2CNRS, Inserm, CHU Lille, Institut Pasteur de Lille, U1019-UMR9017-CIIL-Centre d’Infection et d’Immunité de Lille, Pôle de Médecine Intensive-Réanimation, Université de Lille, 59000 Lille, France; 3Laboratoire de Microbiologie, CHRU de Lille, 2 Avenue Oscar Lambret, 59000 Lille, France

**Keywords:** drug poisoning, aspiration pneumonia, pneumonitis, antibiotics, mechanical ventilation

## Abstract

Drug poisoning frequently leads to admission to intensive care units, often resulting in aspiration, a potentially life-threatening condition if not properly managed. Aspiration can manifest as either bacterial aspiration pneumonia (BAP) or aspiration pneumonitis (AP), which are challenging to distinguish potentially leading to overprescription of antibiotics and the emergence of multidrug-resistant bacteria. This study aims to assess the accuracy of the Infectious Diseases Society of America (IDSA) and British Thoracic Society (BTS) criteria in differentiating BAP from AP in comatose ventilated patients following drug poisoning. This cross-sectional study included 95 patients admitted for drug poisoning at the Lille University Hospital intensive care department, between 2013 and 2017, requiring mechanical ventilation and receiving antibiotics for aspiration. Patients were categorized as having bacterial complications if tracheal sampling yielded positive culture results, and if they were otherwise considered to have chemical complications. The sensitivity, specificity, positive predictive value, and negative predictive value of IDSA and BTS criteria in identifying patients with bacterial complications were evaluated. Among the patients, 34 (36%) experienced BAP. The IDSA criteria demonstrated a sensitivity of 62% and specificity of 33%, while the BTS criteria showed a sensitivity of 50% and specificity of 38%. Both the IDSA and BTS criteria exhibited poor sensitivity and specificity in identifying microbiologically confirmed pneumonia in comatose ventilated patients following drug poisoning.

## 1. Introduction

Acute poisoning is a major public health issue and a common cause of hospitalization. It can occur in different contexts, with accidental causes which may be linked to an acute pathology such as acute kidney failure or to the misuse of alternative medicine [1], or intentional causes, particularly with suicidal intent.

Hospital admission due to suicide attempts is a common occurrence, particularly in France where a report spanning from 2004 to 2011 revealed that between 73,000 to 90,000 patients were hospitalized annually following such attempts. Most of these patients were women aged between 15 to 49 years old. Drug poisoning stands out as the primary method, accounting for 82% of hospital admissions [2]. While the overall mortality rate is relatively low, at 0.8%, it can increase to 10% in cases of overdose involving cardiotropic drugs [3].

An important factor contributing to admission to the intensive care unit (ICU) is the loss of consciousness, which may escalate to a coma, necessitating mechanical ventilation. Neurological conditions such as a coma are recognized as significant risk factors for aspiration complications [4], which in turn can precipitate acute respiratory distress syndrome (ARDS), impacting patient outcomes.

Aspiration pneumonia frequently shows up in patients diagnosed with community-acquired pneumonia (CAP), affecting an estimated 5 to 15% of cases according to various studies [5,6]. However, in some studies, this condition may reach as high as 23%, with an associated mortality rate of approximately 10% [7]. Furthermore, aspiration increases the risk of death by two to three times, and is associated with prolonged hospital stays and an increased likelihood of ICU admission [5,8]. 

Aspiration encompasses two distinct entities, as delineated by Marik et al., which exhibit similar clinical presentations but diverge in their underlying pathophysiology: bacterial aspiration pneumonia (BAP), resulting from the microaspiration of colonized oropharyngeal secretions, and aspiration pneumonitis (AP), caused by the inhalation of noninfectious gastric contents [9]. This latter one, unlike BAP, usually recovers spontaneously within 36 h without requiring antimicrobial therapy [10].

Differentiating these two clinical features may be challenging, with poor accuracy of the biological markers such as C-Reactive Protein, amylasis and Procalcitonin [11,12,13,14]. 

Consequently, prophylactic antimicrobial therapy is frequently initiated. However, Dragan et al. have shown that this approach not only lacks efficacy, in terms of improving survival, but may also be detrimental, potentially leading to increased antibiotic escalation [15] and infectious complications [16].

A positive result for microbiological cultures obtained from endotracheal sampling is necessary for a conclusive diagnosis. However, Rebuck et al. reported that 52% of patients were administered antibiotics when aspiration was suspected, even in the absence of confirmed infection [17]. Further, Lascarrou et al. showed that antibiotics could be interrupted without risks in AP [18].

The misuse of antibiotics contributes to the emergence of multidrug-resistant (MDR) bacteria, posing a significant public health concern. Consequently, distinguishing AP from BAP remains crucial in the management of critically ill patients. 

Few studies have investigated the specific population of patients with drug overdose. Among these patients admitted to the ICU, AP occurred in 11 to 17%, with a mortality rate reaching 4.3% [19,20]. Lascarrou et al. found a similar incidence for BAP [18].

None of these studies have focused on differentiating AP from BAP in comatose patients requiring mechanical ventilation following drug poisoning. More specifically, no study to date has evaluated the accuracy of the usual pneumonia criteria, as defined by the international recommendations of the Infectious Diseases Society of America (IDSA) and the aspiration pneumonia criteria from the British Thoracic Society (BTS), to differentiate between microbiologically confirmed BAP and AP.

This study was conducted to assess the diagnostic accuracy of both BTS and IDSA criteria in distinguishing between microbiologically confirmed BAP and AP within this particular patient population.

## 2. Results

### 2.1. Characteristics of the Study Population

The main characteristics of the study population are outlined in Table 1. During the study period, we enrolled a total of 95 patients. Thirty-four patients (36%) received a diagnosis of BAP based on positive tracheal sampling cultures, while 61 (64%) patients were diagnosed with AP. 

The median age of the cohort was 47 years, with the majority being female (61%). The median SAPS II score was 56, and the median SOFA score was 7. 

During prehospital care, the median Glasgow coma scale was 5. Sixty-one patients (64%) required mechanical ventilation during this phase, with 23 (25%) of them due to acute respiratory failure. Acute poisoning was intentional in all patients.

We found no difference between the AP and BAP groups in baseline characteristics.

Regarding drug poisoning, benzodiazepines were the most commonly used drugs, accounting for 67% of cases, while mixed drug abuse was observed in 74% of cases. Details of the medications are presented in Appendix A. 

The distribution of pathogens identified in tracheal samples is shown in Appendix A. Out of the 34 positive sampling cultures, *Methicillin-susceptible Staphylococcus aureus* was the most identified bacterium (34%), followed by *Haemophilus influenzae* (32%) and *Streptococcus pneumoniae* (26%). Other pathogens notably included *Enterobacteriaceae*.

### 2.2. Criteria for Pneumonia

Upon admission to the ICU, the IDSA criteria for pneumonia were met in 21 (62%) patients with BAP and 41 (67%) patients with AP (*p* = 0.76). In other terms, for the diagnosis of pneumonia, the IDSA criteria had a sensitivity of 62%, a specificity of 33%, a positive predictive value (PPV) of 34%, and a negative predictive value (NPV) of 61% in our cohort.

In addition, the BTS criteria for aspiration pneumonia were met in 17 (50%) patients with BAP and 38 (62%) patients with AP (*p* = 0.34). This means that, for the diagnosis of pneumonia, the BTS criteria had a sensitivity of 50%, a specificity of 38%, a positive predictive value (PPV) of 31%, and a negative predictive value (NPV) of 58% in our cohort.

There were no significant differences between the two groups regarding the variables related to the features of pneumonia (Table 2). 

### 2.3. Outcomes

Ninety-one patients (96%) were discharged alive from the ICU in our cohort. The median duration of invasive mechanical ventilation was 3 days, and the median length of stay in the ICU was 6 days. Patients underwent vasopressors for a median duration of 0 day. MDR bacteria colonization occurred in 14 (15%) patients during the stay in ICU, and was found in 5 (36%) patients prior to the initiation of antibiotic treatments.

Patients’ outcomes were not different between groups (Table 3).

## 3. Discussion

To our knowledge, this study is the first one to evaluate the diagnostic value of the BTS and IDSA criteria in differentiating AP and BAP in the specific population of comatose patients requiring mechanical ventilation after drug poisoning. This population is noteworthy due to its younger age and lower comorbidity burden compared to the typical study population reported in this field. It is noteworthy that the predominant bacteria isolated in our study align with findings reported in the existing literature [7,18,21,22,23]. 

Mortality reached 4% in our study, with no difference between the two groups. This is consistent with the previous literature concerning patients with overdose [19]. However, assessing the mortality rate specifically attributed to aspiration seems challenging. The observed mortality rate is lower than the typical rates seen in cases of aspiration pneumonia. This difference may be attributed to the characteristics of the patients in this population, who are younger and have fewer comorbidities. 

In our investigation, the incidence of BAP was 36% among patients suspected of having BAP, with AP accounting for 64% of the overall cases. This closely aligns with the findings reported by Lascarrou et al., where 44% of patients suspected of having BAP were confirmed with this diagnosis [18]. These results highlight once again the fact that AP is more common than BAP and underscore the need for reliable tests to limit antibiotic overprescription.

MDR bacterial carriage was observed in 15% of patients with pneumonitis. Among them, 78% acquired this carriage during their ICU stay, likely due to unnecessary antibiotic administration following negative tracheal samplings. This underscores the urgency for rapid and accurate tests to facilitate antibiotic discontinuation. Polymerase chain reaction (PCR) could prove to be a valuable tool in this regard. Ongoing studies on this topic may offer insights into addressing this issue.

Our results show the poor performances of both the IDSA and BTS criteria for differentiating BAP from AP. Further, none of the criteria used in both classifications seemed accurate for the prediction of microbiologically confirmed pneumonia, underlining the limited relevance of using these definitions to decide whether to start antimicrobial therapy or not. These results seem consistent with previous data published by our group, highlighting the poor accuracy of the criteria used to differentiate BAP from AP in patients with status epilepticus [24].

Our study presents several limitations. Firstly, the cross-sectional aspect exposes us to missing data. Secondly, the small number of included patients, i.e., 95 subjects from 2013 to 2017, is also a limitation of our study, affecting the power of our statistical analysis, and therefore limiting the validity of our results. Thirdly, the single-center design limits the generalizability of our findings to other centers. Fourthly, a positive X-ray within the first 48 h was considered as positive for the diagnosis in our study, even if the admission X-ray was negative. Indeed, previous data found that X-ray could be negative in aspiration complications [25]. Fifthly, concerning microbiological sampling, 35% of all tracheal sputum were collected after antibiotics initiation in the ICU, mostly in patients finally diagnosed with AP. This could have led to pneumonia misdiagnosing, even though the comparison between groups remained non-statistically significant. Sixthly, we did not collect the mode of respiratory support before intubation for the 36% of patients who were intubated after admission to the ICU, which could have helped us to better understand the typology of patients included in our study. Similarly, information on mechanical ventilation weaning methods, the reintubation rate, or the use of non-invasive ventilation after extubation was not collected. Seventhly, all patients included in our cohort were admitted to the hospital for intentional poisoning, thus limiting the generalizability of our results to all contexts of acute poisoning.

## 4. Materials and Methods

### 4.1. Study Design

We conducted a single-center, cross-sectional study at the intensive care department of Lille University Hospital, including patients admitted between 2013 and 2017.

The main objective of this study was to evaluate, in comatose ventilated patients after drug poisoning, the respective accuracies of the IDSA and BTS criteria for pneumonia in distinguishing microbiologically confirmed BAP from AP. 

The secondary objective of this study was to evaluate outcomes in the BAP and AP groups.

### 4.2. Population

Patients were identified by matching data from two sources: a coding database and hospitalization software of the ICU department (Intelligence space and critical care anesthesia, ICCA^®^, Phillips, Amsterdam, The Netherlands).

Patients were included if they met all the following inclusion criteria:-Age ≥ 18 years;-Admitted to ICU for coma following drug poisoning;-Requiring orotracheal intubation and mechanical support.-Receiving antibiotics for aspiration suspicion with endotracheal microbiological sampling;-New lung infiltrate on a chest X-ray occurring within 48 h following ICU admission.

Exclusion criteria were as follows:
-Age < 18 years;-Degenerative neurologic disease;-Antibiotics before ICU admission;-Absence of chest X-ray within 48 h following ICU admission.

### 4.3. Definitions

The BAP and AP groups were defined according to the result of the culture of endotracheal microbiological sample as follows:The BAP group was defined as patients with a positive culture of endotracheal microbiological sample, with the exception of oropharyngeal flora.The AP group was defined as patients with either:
○A negative culture of endotracheal microbiological sample;○Or a culture of endotracheal microbiological sample that is positive for oropharyngeal flora.

For all patients, the microbiological examination consisted of a direct bacteriological examination with culture of the tracheal sample. No rapid tests were performed on the microbiological samples from patients included in this study.

### 4.4. Criteria for Pneumonia

#### 4.4.1. Criteria of the IDSA for the Diagnosis of Pneumonia

We assessed in this study the accuracy of the IDSA criteria for pneumonia published in 2019 [26] to distinguish BAP from AP. These criteria are defined as follows: -Criterion 1: infiltrate on chest imaging;-Criterion 2: at least one respiratory symptom among:
○New or increased cough;○New or increased sputum production;○Dyspnea;○Pleuritic chest pain,-Criterion 3: at least one other sign among:
○Abnormal lung sounds;○Fever;○Leukocytosis;○Hypoxia (SpO2 < 90%).

According to the IDSA guidelines, the clinical presentation was considered to be in favor of:-BAP in subjects matching criteria 1, 2, and 3.-AP in subjects with at least one criterion missing among criteria 1, 2, and 3.

#### 4.4.2. Criteria of the BTS for the Diagnosis of Aspiration Pneumonia

We assessed in this study the accuracy of the BTS criteria for aspiration pneumonia published in 2022 [6] to distinguish BAP from AP. These criteria are defined as follows: -Criterion 1: history of acute respiratory illness (breathlessness, cough, sputum, fever, sweats, and anorexia);-Criterion 2: factors associated with increased risk of micro aspiration;-Criterion 3: radiological evidence of consolidation.

According to the BTS guidelines, the clinical presentation was considered to be in favor of:-BAP in subjects matching criteria 1, 2, and 3.-AP in subjects with at least one criterion missing among criteria 1, 2, and 3.

### 4.5. Data Collection

The following data were collected for this study: -Demographics and pre-existing conditions;-Characteristics of drug poisoning;-Clinical presentation prior to ICU admission;-Clinical presentation on ICU admission;-Microbiological data: endotracheal sampling or blood culture, timing of sampling, and identification of bacteria;-Antibiotic treatments administered during the stay in ICU;-Criteria for the diagnosis of pneumonia according to the IDSA and the BTS;-Outcomes, including duration of invasive mechanical ventilation, length of stay and mortality at ICU discharge, and colonization with multidrug MDR bacteria.

### 4.6. Ethical Aspects

Given the retrospective nature of this study, written consent was waived, in accordance with French law. This study has been declared to the Commission Nationale Informatique et Libertés (CNIL), in accordance with French regulations concerning research on health databases.

### 4.7. Statistical Analysis

Categorical variables were expressed as numbers (percentages). Normally distributed continuous variables were expressed as means (SD). Non-normally distributed variables were expressed as medians (IQR). Normality of the distribution was checked graphically and by using the Shapiro–Wilk test. The association between the two variables was evaluated with a bivariate analysis. All statistical tests were two-tailed, and *p*-values < 0.05 were considered statistically significant. The statistical analyses were performed using R version 4.1.2 (R Foundation for Statistical Computing, Vienna, Austria).

## 5. Conclusions

In comatose patients requiring mechanical ventilation following drug poisoning, both the IDSA and BTS criteria demonstrated poor performance in distinguishing between pneumonia and pneumonitis in our cohort, thus exposing patients to unnecessary antibiotic administration and potential adverse outcomes. If they were confirmed in a prospective multi-center study, these results would emphasize the urgent need for reliable and rapid diagnostic tests to differentiate these clinical features. 

## Figures and Tables

**Table 1 antibiotics-13-00495-t001:** Main characteristics of the population. BMI: body mass index; COPD: chronic obstructive pulmonary disease, SAPS II: simplified acute physiology score II; SOFA: sequential organ failure assessment.

	Overall Population (*n* = 95)	Aspiration Pneumonitis (*n* = 61)	Bacterial Aspiration Pneumonia (*n* = 34)	*p*
Demographics				
Age (years), median (IQR)	47 (33–60)	48 (34–61)	42 (34–52)	0.20
Gender (female), *n* (%)	58 (61)	38 (62)	20 (59)	0.91
Comorbidities				
BMI > 30 kg/m^2^, *n* (%)	22 (31)	12 (27)	10 (37)	0.51
Diabetes, *n* (%)	9 (9)	6 (9)	3 (9)	1
COPD, *n* (%)	8 (8)	4 (7)	4 (12)	0.62
Chronic cardiac disease, *n* (%)	12 (13)	10 (16)	2 (6)	0.25
Chronic kidney disease, *n* (%)	2 (2)	1 (2)	1 (3)	1
Chronic liver disease, *n* (%)	6 (6)	2 (3)	4 (12)	0.23
Chronic alcohol abuse, *n* (%)	31 (33)	19 (31)	12 (35)	0.85
Toxicomania, *n* (%)	14 (15)	10 (16)	4 (12)	0.85
Tabagism, *n* (%)	29 (30)	18 (29)	11 (32)	0.85
Severity on ICU admission				
SAPS II, median (IQR)	56 (44–65)	53 (44–63)	57 (46–69)	0.2
SOFA score, median (IQR)	7 (5–8)	6 (4–8)	7 (5–9)	0.26
Shock, *n* (%)	32 (34)	21 (34)	11 (32)	1
GCS, median (IQR)	4 (3–7)	3 (3–8)	4 (3–7)	0.95
PaO_2_/FiO_2_, median (IQR)	242 (159–353)	232 (167–354)	250 (130–345)	0.62
Prehospital phase				
Intentional poisoning	95 (100)	61 (100)	34 (100)	1
Loss of sight > 6 h, *n* (%)	30 (45)	16 (36)	14 (61)	0.10
Glasgow scale, median (IQR)	5 (3–15)	5 (3–9)	6 (3–13)	0.32
Vomiting, *n* (%)	17 (18)	13 (22)	4 (12)	0.36
Acute respiratory failure, *n* (%)	23 (25)	15 (25)	8 (25)	1
Out of hospital tracheal intubation, *n* (%)	61 (64)	41 (67)	20 (59)	0.55

**Table 2 antibiotics-13-00495-t002:** Characteristics related to the diagnosis of pneumonia. BTS: British Thoracic Society; CRP: C-reactive Protein; ICU: Intensive care unit; IDSA: Infectious Diseases Society of America; IQR: Interquartile range.

	Overall Population (*n* = 95)	Aspiration Pneumonitis (*n* = 61)	Bacterial Aspiration Pneumonia (*n* = 34)	*p*
Criteria for pneumonia on ICU admission				
Pneumonia according to the IDSA criteria, *n* (%)	62 (65)	41 (67)	21 (62)	0.76
Aspiration pneumonia according to the BTS criteria, *n* (%)	55 (58)	38 (62)	17 (50)	0.34
Hypothermia, *n* (%)	34 (36)	24 (39)	10 (29)	0.46
Fever, *n* (%)	67 (71)	14 (23)	9 (27)	0.89
Purulent sputum, *n* (%)	67 (71)	45 (74)	22 (67)	0.63
Abnormal auscultation, *n* (%)	41 (44)	23 (38)	18 (55)	0.2
CRP (mg/L), median (IQR)	9 (1–27.5)	2.5 (1–15)	11 (1–74)	0.07
Procalcitonin (ng/mL), median (IQR)	0.26 (0.1–1.5)	0.12 (0.1–1.9)	0.46 (0.1–1.4)	0.54
Prior to first antibiotic administration				
Fever, *n* (%)	54 (57)	34 (57)	20 (59)	1
Purulent sputum, *n* (%)	48 (51)	29 (48)	19 (56)	0.63
Hyperleukocytosis, *n* (%)	61 (73)	38 (72)	23 (74)	1
Timing of collection of microbiological tracheal sample				
Before first antibiotic administration, *n* (%)	62 (65)	37 (61)	25 (73)	
After first antibiotic administration, *n* (%)	33 (35)	24 (39)	9 (27)	0.3

**Table 3 antibiotics-13-00495-t003:** Patients’ outcomes. ICU: Intensive care unit; IQR: Interquartile range; MDR: Multidrug resistant.

	Overall Population (*n* = 95)	Aspiration Pneumonitis (*n* = 61)	Bacterial Aspiration Pneumonia (*n* = 34)	*p*
Mortality on ICU discharge, *n* (%)	4 (4)	2 (3)	2 (6)	0.94
Mechanical ventilation duration (days), median (IQR)	3 (2–6)	2 (1–5)	3 (2–6)	0.37
ICU length of stay (days), median (IQR)	6 (4–10)	5 (4–10)	6 (4–10)	0.61
Vasopressors use (days), median (IQR)	0 (0–1)	0 (0–1)	0 (0–1)	0.66
MDR bacteria colonization				
Overall ICU stay, *n* (%)	14 (15)	9 (15)	5 (15)	1
Prior to antibiotics, *n* (%)	5 (36)	2 (22)	3 (60)	0.41

## Data Availability

Dataset available on request from the authors.

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
