# Peer review of "Accuracy of the Infectious Diseases Society of America and British Thoracic Society Criteria for Acute Pneumonia in Differentiating Chemical and Bacterial Complications of Aspiration in Comatose Ventilated Patients Following Drug Poisoning"

_antibiotics, 2024, doi:10.3390/antibiotics13060495_

Round 1

Reviewer 1 Report

Comments and Suggestions for Authors

Delforge and co-authors performed a retrospective analysis of clinical records of patients admitted in their ICU for aspiration pneumonia due to intoxication, with the primary aim to analyze the diagnostic accuracy of IDSA and BTS criteria for aspiration pneumonia.

 I compliment the authors on the chosen issue and their efforts.

I believe that the work should be revised in order to solve some issues, that could improve its overall quality.

Major comments:

1)    Among the exclusion criteria, authors listed “Antibiotics before ICU admission”. However, 35% of the collected microbiological samples were performed after antibiotics administration in the ICU. Authors should explain this inconsistency and how to resolve this issue. Should these patients be excluded from the analysis?

2)    64% of the population required tracheal intubation before hospital admission. How was treated the remaining population (NIPPV, HFNC, conventional oxygen therapy, etc)?

3)    What about mechanical ventilation weaning? Rate of re-intubation? NIPPV use after extubation? Do authors have the possibility to introduce these data?

4)    Authors should specify the microbiological tests used: which rapid test on the microbiological sample, if any?

5)    Introduction should be improved and modified: it seems like the whole population analyzed attempted suicide by drug abuse/overdose. However, as drug or substance intoxication is more frequently seen in the general population during acute exacerbation of other illnesses (acute renal failure, dehydration, etc), or in case of herbal or alternative medicine use/misuse (please, cite Bonanno G, Ippolito M, Moscarelli A, et al. Accidental poisoning with Aconitum: Case report and review of the literature. Clin Case Rep. 2020 Feb 5;8(4):696-698. doi: 10.1002/ccr3.2699), authors should restructure the introduction section, broadening their discussion.

Minor comments:

1)    Please, carefully revise the quality of English language.

Comments on the Quality of English Language

Please, carefully revise the quality of English language:

page 3. line 117: The secondary objectives of this study was --> The secondary objectives of this study WERE...

other minor grammar mistakes can be found throughout the manuscript

Reviewer 2 Report

Comments and Suggestions for Authors

Thanks to authors. Both IDSA and BTS criteria perform poorly in distinguishing between pneumonia and pneumonitis. Therefore, rapid and safe diagnostic tests are needed to distinguish pneumonia and pneumonitis. I think that if the study is conducted multi-centered and prospective, it will yield more valuable results.

Best regrads

Reviewer 3 Report

Comments and Suggestions for Authors

Thank you for allowing me to review this work. It is a very interesting study that evaluates the sensitivity and specificity values of IDSA and BTS criteria to differentiate infectious or chemical origin pneumonia in ventilated comatose patients after drug abuse intoxication. The study is well-defined and easy to read and understand. However, I would like to make some annotations to contribute to improving its quality.

  1. Title: It should not contain abbreviations or acronyms. Not all potential readers understand the meaning of IDSA and BTS. In fact, I had to delve into the full text to understand it. It should be modified.

  2. Abstract: It seems correct and well-structured. Similar to what I explained earlier. The acronyms IDSA and BTS should be explained before their use, just as the authors do with acronyms like BAP AP, etc.

  3. Introduction: Well-structured. Well-referenced. The theoretical framework of the study is well-defined and delimited. Nothing to add.

  4. Methodology: 4.1. The study design should be better defined. It is a cross-sectional study, not retrospective. The fact that the study data were collected from a past period does not make the study retrospective. It should be defined as a descriptive cross-sectional design. 4.2. Why was the study period chosen from 2013 to 2017? During that period, a total of 95 patients were studied. This sample size provides little power to the study, so the results may not be valid. It should be better explained in the study limitations. 4.3. The subheading of study objectives should be eliminated. It is repetitive. 4.4. Among the ethical aspects of the study, it is explained only that due to its retrospective nature, informed consent was not requested from the patients. This should not exempt the study from being evaluated and approved by a Research Ethics Committee.

  5. Results: 5.1. Consistent with the main objective. 5.2. I miss not having conducted a more in-depth analysis. Given the collected variables, a logistic regression analysis could have been performed to generate a predictive model of which variables might be associated with the risk of presenting bacterial or chemical pneumonia.

  6. Discussion: Okay. Consistent with the results and the theoretical framework.

  7. Conclusion: Okay.

Round 2

Reviewer 3 Report

Comments and Suggestions for Authors

Dear authors,

Thank you very much for accepting my contributions and for the responses you have provided. The study is now better understood, and I believe it is better structured. From my point of view, it now meets the criteria for publication in this journal. Thank you.

Best regards,